# The usefulness of heart rate variability in adolescents with tic disorder: Focused on interplay with quality of life

Young Eun Mok[1], SuHyuk Chi[1], June Kang[2], Jeong-An Gim[3], Jeong-kyung Ko[1], Moon-Soo Lee[ID][1]*

1 Department of Psychiatry, Korea University Guro Hospital, Seoul, Republic of Korea, 2 Department of Brain and Cognitive Engineering, Korea University, Seoul, Republic of Korea, 3 Department of Medical Science, Soonchunhyang University, Chungnam, Republic of Korea

* npboard@korea.ac.kr

## Abstract

### Background

Tic disorders, characterized by involuntary movements or vocalizations, are influenced by neurological and psychological factors. Although an imbalance in neurotransmitter systems, genetic factors, and environmental influences play a significant role in the expression of tic disorders, the precise mechanisms through which autonomic changes influence tic production are not fully understood yet. This study investigates the relationship between tic disorders and heart rate variability (HRV), a physiological marker of autonomic nervous system function. The study sought to identify correlations among tic symptoms, HRV indices, and perceived quality of life.

### Methods

In a cohort of 69 participants (39 with tic disorders and 30 controls), we assessed tic severity using the Yale Global Tic Severity Scale (YGTSS) and quality of life through the KIDSCREEN-27 instrument. HRV parameters were measured to evaluate autonomic nervous system activity.

### Results

Patients with tic disorders exhibited significant differences in HRV measures compared to the control group, indicating altered autonomic nervous system functioning. Our findings revealed notable differences in HRV, especially lower Low Frequency (LF) power in the patient group, suggesting altered autonomic responses potentially linked to chronic stress. Correlations between HRV metrics (notably SDNN and RMSSD) and various life quality dimensions were observed in the patient group. These results underscore a potential interplay between tic symptoms, autonomic balance, and adolescents' perceived quality of life.

### Conclusion

The study highlights the importance of considering autonomic nervous system functioning in tic disorders, particularly in the context of stress and perceived quality of life. Our

**Data availability statement:** The data underlying the results presented in this study are available upon reasonable request. Due to ethical and privacy restrictions, access to the dataset requires approval from the Institutional Review Board (IRB) of Korea University Guro Hospital. Interested researchers may contact the IRB at kughirb@kumc.or.kr for data access inquiries.

**Funding:** This research was supported by a grant from the Korea Health Technology R&D Project through the Korea Health Industry Development Institute (KHIDI), funded by the Ministry of Health & Welfare, Republic of Korea (grant number: HI21C0012). The funders had no role in study design, data collection and analysis, decision to publish, or preparation of the manuscript.

**Competing interests:** The authors have declared that no competing interests exist.

findings, which provide insights into tic disorders' physiological and psychological aspects, have important implications for developing more holistic treatment approaches that consider tic patients' mental and physical well-being.

## 1 Introduction

Tic disorders encompass a range of neurodevelopmental conditions primarily characterized by involuntary, repetitive movements or vocalizations. These symptoms are rooted in dysfunctions within the basal ganglia pathways, a critical component of the motor control system in the brain [1]. The basal ganglia's role in these disorders highlights the complex interplay between neuroanatomical structures and neurochemical imbalances.

The production of tics is closely associated with an imbalance in neurotransmitter systems, particularly an excess of dopamine within the striatum. This excess leads to the overstimulation of thalamocortical circuits, integral to motor control and sensory processing [2]. The disruption of these circuits is a key factor in the manifestation of tic symptoms. Additionally, genetic factors and environmental influences play a significant role in developing and expressing tic disorders [3,4].

External stressors play a significant role in exacerbating tic disorders. Activation of the hypothalamic-pituitary-adrenal (HPA) axis in response to stress leads to increased dopamine production, further intensifying the dysfunction within tic-producing pathways [5]. This relationship underscores the sensitivity of tic disorders to environmental and psychological stressors. Studies have shown that stress and anxiety can significantly worsen tic symptoms, suggesting a bidirectional relationship between stress and tic severity [6].

The autonomic nervous system (ANS) is also implicated in the expression of tic disorders. Dysregulation of the ANS, particularly during emotionally arousing events, can worsen tic symptoms. This is evidenced by increased sympathetic tone and decreased vagal (parasympathetic) tone during stressful situations [7]. The precise mechanisms through which these autonomic changes influence tic production are not fully understood, highlighting a possible gap in current research.

Heart rate variability (HRV) is a critical physiological marker for assessing the body's response to stress and ability to adapt to environmental changes. HRV measures the variation in time intervals between successive heartbeats, providing insights into the balance of sympathetic and parasympathetic activity within the ANS [8]. Parameters such as the Standard Deviation of NN Intervals (SDNN) and the Root Mean Square of Successive Differences (RMSSD) are used to evaluate overall ANS activity and parasympathetic function, respectively [9,10].

Despite the established role of HRV in assessing stress responses, its relationship with tic disorders in adolescents still needs to be explored. We hypothesize that adolescents with tic disorder may exhibit altered ANS responses, making them more susceptible to chronic stress. This could manifest in distinct HRV patterns, potentially serving as a biomarker for stress-related exacerbation of tic symptoms. Understanding the connection between tic disorder and HRV could provide valuable insights into the mechanisms underlying these disorders and inform more effective management strategies for adolescents.

## 2 Materials and methods

### 2.1 Selection criteria

A total of 39 patients with tic disorder and 30 healthy controls were initially enrolled in the study. All participants were between the ages of 6 and 18, were psychotropic medication-free

**Table 1. Demographic and clinical characteristics.**

|  | Patient group | Control group |
|---|---|---|
| Age (years) | 9.51 ± 2.76 | 9.84 ± 2.30 |
| Sex | 31:8 | 18:12 |
| IQ* | 97.21 ± 10.99 | 103.45 ± 10.78 |
| YGTSS |  |  |
| Motor tic score | 7.15 ± 3.74 | – |
| Phonic tic score | 4.21 ± 4.61 | – |
| Impairment score | 12.56 ± 7.85 | – |
| KIDSCREEN |  |  |
| Dimension 2 | 18.88 ± 3.79 | 19.77 ± 3.85 |
| Dimension 3* | 27.58 ± 5.32 | 30.27 ± 3.88 |
| Dimension 4 | 15.94 ± 3.51 | 17.37 ± 2.44 |
| Dimension 5* | 27.36 ± 4.76 | 30.00 ± 3.74 |
| Dimension 6 | 15.70 ± 3.60 | 17.23 ± 2.85 |
| Total* | 105.45 ± 14.98 | 114.63 ± 13.36 |
| ARS** | 14.24 ± 10.31 | 5.90 ± 4.99 |

IQ: intelligence quotient; YGTSS: Yale Global Tic Severity Scale; Dimension 2: physical well-being; Dimension 3: Psychological well-being; Dimension 4: Social support and peers; Dimension 5: Autonomy and parent relations; Dimension 6: School environment; ARS: ADHD rating scale;

*: p-value < 0.05;

**: p-value < 0.01

for at least three weeks, and had no history of neurologic disorders, including head trauma, tumors, or seizures. Patients were clinically diagnosed with tic disorders based on the 5th edition of the Diagnostic and Statistical Manual of Mental Disorders (DSM-5) by child and adolescent psychiatrists. Patients were recruited from the Department of Psychiatry of the General Hospital. Healthy controls were recruited from local schools and kindergartens. The research processes were approved by the Institutional Review Board (IRB) of the General Hospital. All research methods were performed in accordance with the relevant guidelines and regulations. Written consent was obtained from the parents or legal guardians of all participants.

## 2.2  Clinical measures

Intelligence quotients of all patients and controls were examined using the Korean version of the Wechsler Intelligence Scale for Children fourth edition (K-WISC-IV). Patients were assessed using the Korean version of the Kiddie-Schedule for Affective Disorders and Schizophrenia-Present and Lifetime Version (K-SADS-PL) for psychiatric comorbidities and the Yale Global Tic Severity Scale (YGTSS) for tic disorder symptom severity. Both groups completed KIDSCREEN-27 to measure quality of life. The KIDSCREEN instrument assesses five distinct dimensions: Physical Well-being (Dimension 2), Psychological Well-being (Dimension 3), Peers and Social Support (Dimension 4), Autonomy and Parent Relations (Dimension 5), and School Environment (Dimension 6). The Korean version of the clinical interview and investigator-rated Kiddie-Schedule for Affective Disorders and Schizophrenia for School-Age Children -Present and Lifetime Version were applied to evaluate mood disorder diagnosis in all patients. Also, ADHD Rating Scale (ARS) was conducted to measure ADHD symptoms in both groups.

**Table 2. Heart rate variability measures in the patient and control groups.**

|  | Patient group | Control group |
|---|---|---|
| HR (beats per minute) | 89.05 ± 10.87 | 85.87 ± 9.55 |
| SDNN (ms) | 58.56 ± 23.80 | 57.91 ± 15.11 |
| RMSSD (ms) | 42.22 ± 24.75 | 43.40 ± 13.90 |
| VLF ($10^{-3}$ ms$^2$) | 9.30 ± 4.38 | 8.79 ± 4.33 |
| LF ($10^{-3}$ ms$^2$)* | 14.04 ± 9.41 | 19.77 ± 10.53 |
| HF ($10^{-3}$ ms$^2$) | 14.91 ± 14.66 | 20.79 ± 11.49 |
| VHF ($10^{-3}$ ms$^2$) | 1.01 ± 1.29 | 1.08 ± 1.69 |
| LF/HF (ratio) | 1.63 ± 1.46 | 1.18 ± 0.80 |

HR: heart rate; SDNN: the Standard Deviation of the Normal-to-Normal Interval; RMSSD: the square root pf the mean squared differences of successive normal-to-normal intervals; VLF: Very Low Frequency; LF: Low Frequency; HF: High Frequency; VHF: Very High Frequency; LF/HF: a ratio of low frequency to high frequency;

*: p-value < 0.05

## 2.3 Heart rate variability measures

Five-minute electrocardiograms were recorded for each of the participants, and then the HRV parameters were derived. Participants were asked not to smoke or drink tea, coffee, or caffeine-containing soft drinks for 3 hours before the recordings. Measurements were conducted in a quiet, temperature-controlled room (approximately 22°C) with subdued lighting to reduce external stressors. Upon arrival, each participant underwent a 15-minute acclimatization period to ensure a standardized physiological state. Participants were asked to assume a supine position on a comfortable examination table during this period. This posture was chosen to minimize postural influences on cardiovascular dynamics, ensuring that variations in HRV were attributable to intrinsic autonomic activity rather than positional adjustments. Participants were instructed to lie still, keep their arms comfortably at their sides, and breathe spontaneously without attempting to control their breathing rhythm. Data collection was performed using the SA-2000E (Medi-Core, Seoul, Korea). An ECG signal was obtained at 500 samples/sec sampling rate for 5 minutes. The HRV indices were calculated based on the R-peaks detection algorithm after low-pass filtering and detrending. The recordings were excluded from further analysis in case of HRV calculation failure due to severe noise. Recordings with non-sinus beats over 1% of the total number of beats were also discarded. Premature beats and artifacts were carefully removed automatically and manually by visually inspecting all RR intervals. Measured HRV parameters were grouped into time and frequency domains. In the time domain, The Standard Deviation of the Normal-to-Normal Interval (SDNN) was used to estimate the long-term components of HRV, and the square root of the mean squared differences of successive normal-to-normal intervals (RMSSD) was calculated by statistical time domain measurements. In the frequency domain, we also measured the following frequency bands: Very Low Frequency (VLF) (0.00–0.04 Hz), Low Frequency (LF) (0.05–0.15 Hz), and High Frequency (HF) (0.16–0.40 Hz) components as recommended by the Task Force of the European Society of Cardiology and the North American Society of Pacing and Electrophysiology. Initially, LF and HF power were recorded as absolute values. Subsequently, the ratio of LF to HF (LF/HF) was calculated from the absolute values of LF and HF power as single number estimates that are considered to reflect simultaneous modulating effects on both the sympathetic and vagal systems. Normalized units of LF and HF (LF norm and HF norm, respectively) were also calculated as [absolute power of the components/(TP-VLF)×100].

## 2.4  Statistical analysis

An Analysis of Covariance (ANCOVA) was employed to evaluate the differences in Heart Rate Variability (HRV) measures between the patient and control groups, with age and sex included as covariates in the model. Spearman correlation test was used to investigate the correlation between HRV measures and YGTSS in the patient group, as well as the quality of life and ARS in both groups. Fisher's Z transformation was done when significant differences between correlations among the two groups were found. Statistical analyses were performed using Microsoft Excel and SPSS version 23 software (IBM Corp., Armonk, NY, USA). The significance level was set at $p < 0.05$.

# 3  Results

## 3.1  Group characteristics

A total of 69 participants participated in the study (patient group: n=39; control group: n=30). The mean age did not differ significantly between the two groups. Although the intelligence quotients of patients and controls differed statistically, all participants' IQs (76–119) were within the normal range on a clinical level. Therefore, the two groups were comparable to each other. Additionally, co-morbid disorders were observed in fourteen patients (ADHD, n=13; anxiety disorder, n=1).

## 3.2  Clinical measures

In the comparative analysis, the Yale Global Tic Severity Scale (YGTSS) scores exhibited significant differences between the patient and control groups. Specifically, the scores for motor tics, phonic tics, and overall impairment score were recorded as 7.15±3.74, 4.21±4.61, and 12.56±7.85, respectively, in the patient group. In contrast, the control group demonstrated a score of 0 in all motor tic, phonic tic, and impairment parts. Additionally, an assessment

**Table 3. Correlation between heart rate variability measures and clinical rating scales.**

|  |  | SDNN | RMSSD | VLF | LF | HF | VHF | LF/HF |
|---|---|---|---|---|---|---|---|---|
| Patient group | KID 2 | .354 | .308 | -.488** | -.259 | .008 | -.075 | -.161 |
|  | KID 3 | .234 | .180 | -.087 | .040 | .072 | .091 | .162 |
|  | KID 4 | .553** | .552** | -.198 | -.032 | .219 | .071 | -.193 |
|  | KID 5 | .437* | .455* | -.407* | -.113 | .075 | .027 | -.053 |
|  | KID 6 | .351 | .309 | -.193 | -.180 | -.130 | -.055 | .129 |
|  | KID T | .530** | .497** | -.379* | -.140 | .073 | .024 | -.021 |
|  | K-ARS | -.230 | -.039 | .106 | .094 | .207 | .303 | -.275 |
| Control group | KID 2 | -.099 | .085 | -.147 | .147 | .224 | -.032 | -.039 |
|  | KID 3 | -.409* | -.183 | -.252 | .186 | .130 | -.077 | -.067 |
|  | KID 4 | -.242 | -.081 | .146 | .180 | .053 | .030 | -.074 |
|  | KID 5 | -.432* | -.444* | -.182 | .018 | -.190 | -.362 | .101 |
|  | KID 6 | -.326 | -.269 | -.197 | .030 | -.164 | -.523** | -.024 |
|  | KID T | -.381* | -.228 | -.181 | .138 | .020 | -.242 | -.019 |
|  | K-ARS | .104 | -.049 | -.075 | -.056 | -.367 | -.312 | .361 |

KID 2: KIDSCREEN dimension 2; KID 3: KIDSCREEN dimension 3; KID 4: KIDSCREEN dimension 4; KID 5: KIDSCREEN dimension 5; KID 6: KIDSCREEN dimension 6; KID T: KIDSCREEN total; ARS: ADHD rating scale;

*: p-value < 0.05;

**: p-value < 0.01

using the KIDSCREEN-27 revealed disparities in the quality of life measures between the two groups. The control group's KIDSCREEN scores were consistently higher across all dimensions, indicating a comparatively lower subjective quality of life in the patient group. Furthermore, the Attention-Deficit/Hyperactivity Disorder Rating Scale (ARS) scores were elevated in the patient group, highlighting a greater prevalence of ADHD symptoms among individuals with tic disorders (Table 1).

### 3.3  Between-group comparison of HRV measures at baseline

In the context of gender and age covariate adjustments, there was no observed disparity in heart rate between the case and control groups. Additionally, when considering the variables SDNN and RMSSD, no statistically significant distinctions were discerned between the two groups. While VLF exhibited no statistically significant disparity, LF exhibited a noteworthy discrepancy between the groups, with a statistically significant result (F = 5.038, p = 0.028). Conversely, HF, VHF, and LF/HF ratio displayed no statistically significant distinctions between the case and control groups, suggesting their comparability in this study (Table 2).

### 3.4  Correlation between HRV measures and clinical rating scales

Within the tic group, a notable inverse relationship was observed between the ARS measure and specific dimensions of the KIDSCREEN instrument. Dimension 5 of KIDSCREEN exhibited a negative correlation (r = -0.391, p = 0.025), as did Dimension 6 (r = -0.383, p = 0.028), and the overall KIDSCREEN Total (r = -0.390, p = 0.025). Conversely, the control group did not yield any statistically significant correlations between KIDSCREEN dimensions and ARS measures.

Intriguingly, within the tic group, the KIDSCREEN dimensions also demonstrated statistically significant associations with Heart Rate Variability (HRV) measures, including SDNN, RMSSD, and VLF. For instance, Dimension 2 of KIDSCREEN displayed a negative correlation with VLF (r = -0.488, p < 0.05), while Dimension 4 exhibited positive correlations with both SDNN (r = 0.553, p < 0.001) and RMSSD (r = 0.552, p < 0.001). Dimension 5 of KIDSCREEN displayed positive correlations with RMSSD (r = 0.437, p = 0.014) and SDNN (r = 0.455, p = 0.010), but conversely, a negative correlation with VLF (r = -0.407, p = 0.023). Furthermore, the total KIDSCREEN score exhibited positive correlations with RMSSD (r = 0.537, p = 0.004) and SDNN (r = 0.497, p = 0.004), and a negative correlation with VLF (r = -0.379, p = 0.035).

In contrast, within the control group, only Dimension 3 of KIDSCREEN showed a negative correlation with SDNN (r = -0.409, p = 0.031). Dimension 5 of KIDSCREEN exhibited negative correlations with both SDNN (r = -0.432, p = 0.022) and RMSSD (r = -0.444, p = 0.018), while the overall KIDSCREEN total score displayed a negative correlation with SDNN (r = -0.381, p = 0.045). Notably, in both the tic and control groups, no statistically significant correlations were observed between ARS measures and HRV metrics (Table 3).

### 3.5  Comparison of correlation between HRV measures and clinical scales

Our investigation entailed an in-depth examination of the substantial associations existing between Heart Rate Variability (HRV) parameters and clinical scales among the respective participant cohorts. Subsequently, we conducted supplementary analyses based on the identification of notable correlations.

The correlation observed between Dimension Five of the KIDSCREEN instrument and HRV parameters RMSSD and SDNN displays a statistically significant distinction when compared to the control group. Conversely, the correlation between KIDSCREEN Dimension Five and the Very Low-Frequency (VLF) component does not exhibit statistically significant disparities in correlation between the two cohorts.

Similarly, the correlation between the KIDSCREEN Total Score and HRV parameters SDNN and RMSSD within the tic disorder group significantly deviates from that within the control group. In contrast, the distinctions in the correlation between the KIDSCREEN Total Score and the VLF component within both groups appear to lack statistical significance.

## 4  Discussion

As anticipated, the application of KIDSCREEN scales unveiled marked distinctions between the examined groups, thereby emphasizing the unique attributes inherent to the patient cohort. Particularly noteworthy were the discernible disparities in the Attention Deficit Hyperactivity Disorder Rating Scale (ARS) scores between patients and controls, a phenomenon indicative of the well-established comorbidity of attention-deficit/hyperactivity disorder (ADHD) in tic disorders—a recognized characteristic of the patient population. As expected, ARS was higher in the patient group, and there was a significant correlation between ARS and KIDSCREEN in the patient group. Notably, a negative correlation was observed between ARS scores and several KIDSCREEN dimensions, including dimensions 5, 6, and the total score, exclusively within the patient group. This contrasts starkly with the absence of a similar correlation in the control group, suggesting that the severity of ADHD symptoms detrimentally influences familial relationships, autonomy, academic competence, and overall quality of life within the patient group. Given the high prevalence of ADHD in tic patients, our findings underscore the imperative nature of ADHD screening in this demographic to inform and tailor comprehensive care strategies effectively.

Low Frequency (LF) power is often associated with both sympathetic and parasympathetic nervous system activity [11]. LF is thought to reflect several physiological processes, including baroreflex activity, and can be influenced by various factors such as breathing patterns, mental stress, and physical activity [12–14]. LF, along with other HRV metrics, is used to assess autonomic function and has applications in various contexts. It can provide insights into stress response, cardiovascular health, and overall autonomic balance. Our investigations revealed a discernible reduction in the mean LF power within the patient group, representing a deviation from the typically elevated LF levels observed during acute stress state. This reduction aligns with many previous studies positing that chronic stress may incite an adaptive response, leading to diminished autonomic flexibility [15,16]. The consistently lower LF power observed in tic patients strongly suggests a sustained exposure to chronic stress, providing compelling evidence supporting our proposed theoretical framework. Our findings suggest that tic disorders may be associated with persistent physiological stress, even in the absence of immediate emotional distress. Interestingly, HF power, which reflects parasympathetic activity, did not differ significantly between the tic disorder and control groups. This suggests that the primary autonomic alteration in tic disorders may involve reduced adaptability of the sympathetic nervous system rather than an isolated impairment of vagal function.

Standard Deviation of Normal-to-Normal intervals (SDNN) represents the overall variability in heart rate during the measurement period [17]. A higher SDNN value indicates more significant overall heart rate variability, which is generally considered a marker of a healthy heart and a balanced autonomic nervous system [18]. Lower SDNN values can show reduced heart rate variability, which may be associated with various health issues, including increased stress, poor cardiovascular health, or autonomic dysfunction [8,19]. Root Mean Square of Successive Differences (RMSSD) primarily reflects the parasympathetic influences on the heart rate [10]. A higher RMSSD indicates higher parasympathetic activity, often associated with relaxation and recovery [20].

A lower RMSSD can indicate reduced vagal activity, often associated with stress or poor heart health [21]. Our findings can also be reflected upon such interpretations. A notable

positive correlation between SDNN, RMSSD, and KIDSCREEN's dimension 5 (autonomy and parent relations) was found in the patient group. On the other hand, the control group showed a negative correlation between SDNN, RMSSD, and KIDSCREEN's dimension 5. Our study also identified a positive correlation between KIDSCREEN Total scores and both SDNN and RMSSD within the patient cohort.

These results indicate that adolescents with higher autonomic flexibility report better overall well-being and stronger familial relationships. These findings suggest that more significant heart rate variability is associated with better psychological resilience and social adaptation in tic disorder patients. This supports prior studies suggesting that higher vagal tone, reflected by higher SDNN and RMSSD, is linked to better emotional regulation and adaptive stress responses [21]. Conversely, lower HRV is often associated with increased psychological distress, anxiety, and poorer social functioning, which may be particularly relevant in tic disorders where stress exacerbates symptoms [6].

The control group exhibited a negative correlation between KIDSCREEN Total scores and SDNN, a trend that, while not statistically significant, was similarly observed in the relationship between KIDSCREEN Total scores and RMSSD. This notable variance in correlation coefficients between the patient and control groups underscores the divergent responses to stress exhibited by each group. The findings highlight a marked difference in autonomic responses between individuals with tic disorders and healthy controls. Furthermore, this divergence is evident when assessed through conventional psychophysiological metrics. The observed associations between SDNN, RMSSD, and autonomic regulation emphasize the vital connection between life quality—particularly in familial relationships and personal autonomy—and stress adaptability in individuals with tic disorders.

It is well-known that HRV can serve as a psychophysiological marker [8]. Our findings suggest that employing psychophysiological assessments like HRV can be beneficial in studying tic disorders. HRV stands out due to its brevity, adaptability, and ease of application, making it a potentially vital tool for objectively measuring psychological stress in tic disorders. Regarding the difference in correlations with KIDSCREEN and HRV measures among the patient and control groups, the study indicates distinct stress responses between the patients and control groups. Our finding highlights potential differences reflected by traditional psychophysiological evaluations. Overall, this study supports the effectiveness of using HRV in the context of tics. In a recent study, art was introduced for the purpose of relaxation, and the art intervention showed significantly greater physiological relaxation, indicated by an increase in HRV parameters. This study also shows the effectiveness of HRV for measuring physiological arousal status [22].

There are, however, some limitations to this study. Our study was part of a broader study that mainly focused on patients with tic disorder rather than Tourette Syndrome. Some participants with tics in our study reported minimal functional impairment, as indicated by the YGTSS (Yale Global Tic Severity Scale) score. This may be a limitation to our study in that somewhat less severe tic disorder patients were evaluated. A significant correlation between the YGTSS, KIDSCREEN (quality of life measure), and HRV was not found, and this may be partly affected by lower symptom severity. Although our sample size was based on previous literature and was sufficient for detecting significant differences in HRV measures, a larger cohort with a priori power analysis would enhance the robustness of future studies. This should be considered in future research to ensure optimal statistical power and generalizability. Future research would benefit from assembling a more extensive cohort of patients exhibiting severe tic symptoms, allowing for a separate and detailed analysis of this specific subgroup.

In light of these findings, our study strongly advocates for prioritizing quality of life considerations in the management of tic disorders, positing that such an approach holds promise for significantly enhancing mental and physical health outcomes in this complex patient population.

## 5 Conclusion

Our study presents significant insights into the complex interplay between tic disorders and autonomic nervous system functioning in adolescents, focusing on heart rate variability (HRV) as an indicator of stress response. The findings reveal notable differences in HRV measures between adolescents with tic disorders and the control group, underscoring altered autonomic functioning in the former. A key finding of our study is that adolescents with tic disorders exhibited significantly lower LF power than controls, suggesting an altered autonomic response pattern. Another key result was the positive correlation between SDNN, RMSSD, and various KIDSCREEN dimensions in the patient group, particularly autonomy and parent relations (Dimension 5). This suggests that greater HRV, indicative of better autonomic regulation, is linked to improved quality of life and psychological resilience in tic disorder patients. Conversely, a negative correlation between these HRV indices and quality of life was observed in the control group. This discrepancy underscores a fundamental difference in autonomic regulation between individuals with tic disorders and typically developing controls.

These results collectively highlight the intricate interplay between autonomic function, stress adaptation, and quality of life in tic disorder patients. The observed differences in HRV patterns suggest that tic disorders are not merely a neuropsychiatric condition but also involve physiological dysregulation that warrants further investigation. Future studies should explore whether targeted interventions, such as stress management techniques and HRV-based training, can improve autonomic function and reduce tic burden. In light of these findings, our study strongly advocates for a more integrative approach to tic disorder management, considering both psychological and physiological factors. The observed autonomic dysregulation suggests that incorporating HRV assessments and stress regulation strategies may enhance treatment outcomes and improve adolescents' overall well-being.

## Author contributions

**Conceptualization:** Young Eun Mok, Jeong-An Gim, Jeong-kyung Ko, Moon-Soo Lee.

**Data curation:** Jeong-An Gim, Jeong-kyung Ko.

**Formal analysis:** June Kang, Jeong-kyung Ko.

**Funding acquisition:** Moon-Soo Lee.

**Investigation:** Young Eun Mok, SuHyuk Chi, June Kang, Jeong-An Gim, Moon-Soo Lee.

**Methodology:** SuHyuk Chi, Jeong-An Gim, Moon-Soo Lee.

**Project administration:** Moon-Soo Lee.

**Software:** June Kang.

**Supervision:** SuHyuk Chi, Moon-Soo Lee.

**Validation:** SuHyuk Chi, Jeong-An Gim.

**Visualization:** June Kang.

**Writing – original draft:** Young Eun Mok, SuHyuk Chi, Moon-Soo Lee.

**Writing – review & editing:** SuHyuk Chi, June Kang, Jeong-An Gim, Jeong-kyung Ko, Moon-Soo Lee.

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
