## [Decision Letter · Decision Letter 0]

26 Dec 2024

PONE-D-24-09801The usefulness of heart rate variability in adolescents with tic disorder: Focused on interplay with quality of lifePLOS ONE

Dear Dr. Lee,

Thank you for submitting your manuscript to PLOS ONE. After careful consideration, we feel that it has merit but does not fully meet PLOS ONE’s publication criteria as it currently stands. Therefore, we invite you to submit a revised version of the manuscript that addresses the points raised during the review process.

We look forward to receiving your revised manuscript.

Kind regards,

Metha Chanda, D.V.M.,Ph.D., DTBVM

Academic Editor

PLOS ONE

“This research was supported by a grant from the Korea Health Technology R&D Project through the Korea Health Industry Development Institute (KHIDI), funded by the Ministry of Health & Welfare, Republic of Korea [grant number: HI21C0012].”

3. In the online submission form you indicate that your data is not available for proprietary reasons and have provided a contact point for accessing this data. Please note that your current contact point is a co-author on this manuscript. According to our Data Policy, the contact point must not be an author on the manuscript and must be an institutional contact, ideally not an individual. Please revise your data statement to a non-author institutional point of contact, such as a data access or ethics committee, and send this to us via return email. Please also include contact information for the third party organization, and please include the full citation of where the data can be found.

Reviewers' comments:

Reviewer's Responses to Questions

**Comments to the Author**

1. Is the manuscript technically sound, and do the data support the conclusions?

Reviewer #1: Yes

Reviewer #2: Yes

2. Has the statistical analysis been performed appropriately and rigorously? 

Reviewer #1: Yes

Reviewer #2: Yes

3. Have the authors made all data underlying the findings in their manuscript fully available?

Reviewer #1: Yes

Reviewer #2: Yes

4. Is the manuscript presented in an intelligible fashion and written in standard English?

Reviewer #1: Yes

Reviewer #2: Yes

5. Review Comments to the Author

Reviewer #1: Discussion looks little bit short, could have been little descriptive. It is recommended to that it can be rewrite in little detailed manner. In Methods section study participants were asked to remain stationary and instructed not to move, here can you please elaborate which position preferred to record the values? Please specify it.

Reviewer #2: I would like to appreciate the efforts of the authors in implementing the project and writing this article “The usefulness of heart rate variability in adolescents with tic disorder: Focused on interplay with quality of life”.

This study investigates the relationship between tic disorders and heart rate variability (HRV), a physiological marker of autonomic nervous system function.

The aim of this study is to identify correlations between tic symptoms, HRV indices and perceived quality of life.

I have these comments and questions:

The work brings interesting results, which can be very useful and beneficial for practice. However, I have some comments and questions.

The units should be listed in the tables (if any).

I recommend that the unbalanced sex ratio of the participants should be included in the study limits.

Has the Power Analysis been addressed? If so, what is the optimal number of participants to demonstrate the observed phenomenon? The results are interesting, but it might be appropriate to include the number of participants in the study limits.

The conclusion is written a little more generally, I recommend to specifically highlight the main result of the study.

In the last 2 years, there have been published interesting studies on tic or HRV, which are not included in the introduction or discussion of this article.

6. PLOS authors have the option to publish the peer review history of their article (what does this mean?). If published, this will include your full peer review and any attached files.

Reviewer #1: **Yes: **Nawaj Mehtab Pathan

Reviewer #2: No

---

## [Author Response · Author response to Decision Letter 1]

5 Feb 2025

Dear Editor and Reviewers,

We really would like to thank you for the review and valuable recommendations that were very helpful in improving our manuscript. In line with your advice, I have revised the text accordingly.

Notes: Below, we have provided (1) the reviewer’s comments, (2) our responses, and (3) revisions on an item-by-item basis.

Reviewer #1

1. In the Methods section study participants were asked to remain stationary and instructed not to move, here can you please elaborate which position preferred to record the values? Please specify it.

Response: We have added more specific details about the experiment and posture to get the HRV measures. As the reviewer suggested, this also includes the instructions for immobilization.

Revision:

Participants were asked to remain stationary during recordings and refrain from moving. � “Measurements were conducted in a quiet, temperature-controlled room (approximately 22°C) with subdued lighting to reduce external stressors. Upon arrival, each participant underwent a 15-minute acclimatization period to ensure a standardized physiological state. Participants were asked to assume a supine position on a comfortable examination table during this period. This posture was chosen to minimize postural influences on cardiovascular dynamics, ensuring that variations in HRV were attributable to intrinsic autonomic activity rather than positional adjustments. Participants were instructed to lie still, keep their arms comfortably at their sides, and breathe spontaneously without attempting to control their breathing rhythm.”

2. Discussion looks a little bit short, could have been little descriptive. It is recommended that it can be rewritten in little detailed manner.

Response: We appreciate your recommendation and have added more specificity and richness to the discussion based on our research.

Revision:

“Our findings suggest that tic disorders may be associated with persistent physiological stress, even in the absence of immediate emotional distress. Interestingly, HF power, which reflects parasympathetic activity, did not differ significantly between the tic disorder and control groups. This suggests that the primary autonomic alteration in tic disorders may involve reduced adaptability of the sympathetic nervous system rather than an isolated impairment of vagal function.”

“These results indicate that adolescents with higher autonomic flexibility report better overall well-being and stronger familial relationships. These findings suggest that more significant heart rate variability is associated with better psychological resilience and social adaptation in adolescents with tic disorder. This supports prior studies suggesting that higher vagal tone, reflected by higher SDNN and RMSSD, is linked to better emotional regulation and adaptive stress responses [21]. Conversely, lower HRV is often associated with increased psychological distress, anxiety, and poorer social functioning, which may be particularly relevant in tic disorders where stress exacerbates symptoms [6].”

Reviewer #2

1. The units should be listed in the tables (if any).

Response: We have added the units within the tables as requested. Minor modifications were added to the tables.

Revision:

In table 1, age � age (years)

In table 2, HR (bpm) � HR (beats per minute), LF/HF(N/A) � LF/HF (ratio)

In table 3, ARS � K-ARS, HR (bpm) � HR (beats per minute), LF/HF(N/A) � LF/HF (ratio)

2. I recommend that the unbalanced sex ratio of the participants should be included in the study limits.

Response: We appreciate your detailed recommendation. In this study, the ratio of males to females in the patient group is 31 to 8, and the ratio of males to females in the control group is 18 to 12. The male-female ratio was not statistically significantly different between the patient and control groups. (Chi-square value: 2.253, p-value: 0.133, degrees of freedom: 1) in chi-square test. Here, the p-value is 0.133, which is greater than the usual significance level of 0.05, so the difference in the ratio of males to females between the patient and control groups is not statistically significant. The reviewer's point about our study is that more males than females were observed in both the patient and control groups. However, tics have traditionally been known to be more common in males, and recent studies have primarily supported this trend. So, our findings reflect these real-world research and clinical observations. We had more males in the patient population and more males in the matching control group to ensure comparability. Therefore, this does not need to be included in the study limitations.

3. Has the Power Analysis been addressed? If so, what is the optimal number of participants to demonstrate the observed phenomenon? The results are interesting, but it might be appropriate to include the number of participants in the study limits.

Response: We appreciate the valuable feedback regarding power analysis and sample size considerations, which are required to ensure the robustness of our findings. While a formal a priori power analysis was not conducted prior to participant recruitment, we carefully determined our sample size based on previous studies examining heart rate variability (HRV) in tic disorder populations. Prior research on similar topics has used comparable or smaller sample sizes (Tic Frequency Decreases during Short-term Psychosocial Stress – An Experimental Study on Children with Tic Disorders, Front. Psychiatry, 17 May 2016 Sec. Child and Adolescent Psychiatry – 31 children and adolescents with tic disorders/ Autonomic cardiovascular regulation in patients with tics and Tourette syndrome, Zh Nevrol Psikhiatr Im S S Korsakova. 2005;105(9):18-22. - patients aged 4-15 years with Tourette syndrome (n = 22) and other tic disorders (n = 48) ), supporting the feasibility of our participant count. To address the concern, we conducted a post-hoc power analysis using the effect sizes derived from our study results. Our calculations indicate that the required number of participants was calculated to be 70 when we use the LF index (key parameter of our statistical analysis) for calculation (actual group means, alpha 0.05, beta 0.2, power 0.8, calculated from sample size calculator), suggesting that our sample size was sufficient to detect meaningful differences. However, we acknowledge that larger sample sizes could enhance the generalizability of our findings and reduce the risk of Type II errors. We recognize that a larger cohort would strengthen the study’s conclusions. As noted in our limitations section, future research should aim to include a broader sample and conduct a priori power analysis to further refine participant recruitment strategies. We have clarified this point in the revised manuscript by explicitly discussing the sample size as a potential study limitation.

Revision: To incorporate this valuable feedback, we have added the following statement to the Discussion section:

" Although our sample size was based on previous literature and was sufficient for detecting significant differences in HRV measures, a larger cohort with a priori power analysis would enhance the robustness of future studies. This should be considered in future research to ensure optimal statistical power and generalizability."

4. The conclusion is written a little more generally, I recommend specifically highlighting the main result of the study. In the last 2 years, there have been published interesting studies on tic or HRV, which are not included in the introduction or discussion of this article.

Response: We added more specific texts to the conclusion and highlighted the study's main result. We have also added more related studies on tic studies using HRV in the discussion section.

Revision:

We have added following texts in the discussion section.

In a recent study, art was introduced for the purpose of relaxation, and the art intervention showed significantly greater physiological relaxation, indicated by an increase in HRV parameters. This study also shows the effectiveness of HRV for measuring physiological arousal status [22].

We have rewritten the conclusion section as follows:

Specifically, the observed lower Low Frequency (LF) power in patients with tic disorders suggests a potential chronic stress state, aligning with previous research that indicates chronic stress can lead to diminished autonomic flexibility. � A key finding of our study is that adolescents with tic disorders exhibited significantly lower LF power than the controls, suggesting an altered autonomic response pattern. Another key result was the positive correlation between SDNN, RMSSD, and various KIDSCREEN dimensions in the patient group, particularly autonomy and parent relations (Dimension 5). This suggests that greater HRV, indicative of better autonomic regulation, is linked to improved quality of life and psychological resilience in tic disorder patients. Conversely, a negative correlation between these HRV indices and quality of life was observed in the control group. This discrepancy underscores a fundamental difference in autonomic regulation between individuals with tic disorders and typically developing controls.

The correlations between HRV parameters, particularly SDNN and RMSSD, and various dimensions of quality of life in the patient group highlight the intricate relationship between physiological stress responses, tic symptoms, and overall quality of life. Our study advocates for integrating stress management strategies and interventions to improve autonomic regulation in treating tic disorders. Such approaches could potentially enhance the quality of life for adolescents suffering from these conditions.

These results collectively highlight the intricate interplay between autonomic function, stress adaptation, and quality of life in tic disorder patients. The observed differences in HRV patterns suggest that tic disorders are not merely a neuropsychiatric condition but also involve physiological dysregulation that warrants further investigation. Future studies should explore whether targeted interventions, such as stress management techniques and HRV-based training, can improve autonomic function and reduce tic burden. In light of these findings, our study strongly advocates for a more integrative approach to tic disorder management, considering both psychological and physiological factors. The observed autonomic dysregulation suggests that incorporating HRV assessments and stress regulation strategies may enhance treatment outcomes and improve adolescents' overall well-being.

In conclusion, this research contributes to a deeper understanding of the psychophysiological aspects of tic disorders. It highlights the critical role of stress and autonomic nervous system functioning in these conditions. Future studies should aim to explore the long-term effects of stress management and autonomic regulation interventions on tic disorders, potentially paving the way for more effective and comprehensive treatment modalities.

---

## [Editor Report · Decision Letter 1]

9 Feb 2025

PONE-D-24-09801R1The usefulness of heart rate variability in adolescents with tic disorder: Focused on interplay with quality of lifePLOS ONE

Dear Dr. Lee,

Thank you for submitting your manuscript to PLOS ONE. After careful consideration, we feel that it has merit but does not fully meet PLOS ONE’s publication criteria as it currently stands. Therefore, we invite you to submit a revised version of the manuscript that addresses the points raised during the review process.

We look forward to receiving your revised manuscript.

Kind regards,

Metha Chanda, D.V.M.,Ph.D., DTBVM

Academic Editor

PLOS ONE
---

## [Author Response · Author response to Decision Letter 2]

14 Feb 2025

Dear Editor and Reviewers,

We really would like to thank you for the review and valuable recommendations that were very helpful in improving our manuscript. In line with your advice, I have revised the text accordingly.

Notes: Below, we have provided (1) the reviewer’s comments, (2) our responses, and (3) revisions on an item-by-item basis.

Reviewer #1

1. In the Methods section study participants were asked to remain stationary and instructed not to move, here can you please elaborate which position preferred to record the values? Please specify it.

Response: We have added more specific details about the experiment and posture to get the HRV measures. As the reviewer suggested, this also includes the instructions for immobilization.

Revision:

[Participants were asked to remain stationary during recordings and refrain from moving.] -> “Measurements were conducted in a quiet, temperature-controlled room (approximately 22°C) with subdued lighting to reduce external stressors. Upon arrival, each participant underwent a 15-minute acclimatization period to ensure a standardized physiological state. Participants were asked to assume a supine position on a comfortable examination table during this period. This posture was chosen to minimize postural influences on cardiovascular dynamics, ensuring that variations in HRV were attributable to intrinsic autonomic activity rather than positional adjustments. Participants were instructed to lie still, keep their arms comfortably at their sides, and breathe spontaneously without attempting to control their breathing rhythm.”

2. Discussion looks a little bit short, could have been little descriptive. It is recommended that it can be rewritten in little detailed manner.

Response: We appreciate your recommendation and have added more specificity and richness to the discussion based on our research.

Revision:

“Our findings suggest that tic disorders may be associated with persistent physiological stress, even in the absence of immediate emotional distress. Interestingly, HF power, which reflects parasympathetic activity, did not differ significantly between the tic disorder and control groups. This suggests that the primary autonomic alteration in tic disorders may involve reduced adaptability of the sympathetic nervous system rather than an isolated impairment of vagal function.”

“These results indicate that adolescents with higher autonomic flexibility report better overall well-being and stronger familial relationships. These findings suggest that more significant heart rate variability is associated with better psychological resilience and social adaptation in adolescents with tic disorder. This supports prior studies suggesting that higher vagal tone, reflected by higher SDNN and RMSSD, is linked to better emotional regulation and adaptive stress responses [21]. Conversely, lower HRV is often associated with increased psychological distress, anxiety, and poorer social functioning, which may be particularly relevant in tic disorders where stress exacerbates symptoms [6].”

Reviewer #2

1. The units should be listed in the tables (if any).

Response: We have added the units within the tables as requested. Minor modifications were added to the tables.

Revision:

In table 1, age -> age (years)

In table 2, HR (bpm) -> HR (beats per minute), LF/HF(N/A) -> LF/HF (ratio)

In table 3, ARS -> K-ARS, HR (bpm) -> HR (beats per minute), LF/HF(N/A) -> LF/HF (ratio)

2. I recommend that the unbalanced sex ratio of the participants should be included in the study limits.

Response: We appreciate your detailed recommendation. In this study, the ratio of males to females in the patient group is 31 to 8, and the ratio of males to females in the control group is 18 to 12. The male-female ratio was not statistically significantly different between the patient and control groups. (Chi-square value: 2.253, p-value: 0.133, degrees of freedom: 1) in chi-square test. Here, the p-value is 0.133, which is greater than the usual significance level of 0.05, so the difference in the ratio of males to females between the patient and control groups is not statistically significant. The reviewer's point about our study is that more males than females were observed in both the patient and control groups. However, tics have traditionally been known to be more common in males, and recent studies have primarily supported this trend. So, our findings reflect these real-world research and clinical observations. We had more males in the patient population and more males in the matching control group to ensure comparability. Therefore, this does not need to be included in the study limitations.

3. Has the Power Analysis been addressed? If so, what is the optimal number of participants to demonstrate the observed phenomenon? The results are interesting, but it might be appropriate to include the number of participants in the study limits.

Response: We appreciate the valuable feedback regarding power analysis and sample size considerations, which are required to ensure the robustness of our findings. While a formal a priori power analysis was not conducted prior to participant recruitment, we carefully determined our sample size based on previous studies examining heart rate variability (HRV) in tic disorder populations. Prior research on similar topics has used comparable or smaller sample sizes (Tic Frequency Decreases during Short-term Psychosocial Stress – An Experimental Study on Children with Tic Disorders, Front. Psychiatry, 17 May 2016 Sec. Child and Adolescent Psychiatry – 31 children and adolescents with tic disorders/ Autonomic cardiovascular regulation in patients with tics and Tourette syndrome, Zh Nevrol Psikhiatr Im S S Korsakova. 2005;105(9):18-22. - patients aged 4-15 years with Tourette syndrome (n = 22) and other tic disorders (n = 48) ), supporting the feasibility of our participant count. To address the concern, we conducted a post-hoc power analysis using the effect sizes derived from our study results. Our calculations indicate that the required number of participants was calculated to be 70 when we use the LF index (key parameter of our statistical analysis) for calculation (actual group means, alpha 0.05, beta 0.2, power 0.8, calculated from sample size calculator), suggesting that our sample size was sufficient to detect meaningful differences. However, we acknowledge that larger sample sizes could enhance the generalizability of our findings and reduce the risk of Type II errors. We recognize that a larger cohort would strengthen the study’s conclusions. As noted in our limitations section, future research should aim to include a broader sample and conduct a priori power analysis to further refine participant recruitment strategies. We have clarified this point in the revised manuscript by explicitly discussing the sample size as a potential study limitation.

Revision: To incorporate this valuable feedback, we have added the following statement to the Discussion section:

" Although our sample size was based on previous literature and was sufficient for detecting significant differences in HRV measures, a larger cohort with a priori power analysis would enhance the robustness of future studies. This should be considered in future research to ensure optimal statistical power and generalizability."

4. The conclusion is written a little more generally, I recommend specifically highlighting the main result of the study. In the last 2 years, there have been published interesting studies on tic or HRV, which are not included in the introduction or discussion of this article.

Response: We added more specific texts to the conclusion and highlighted the study's main result. We have also added more related studies on tic studies using HRV in the discussion section.

Revision:

We have added following texts in the discussion section.

In a recent study, art was introduced for the purpose of relaxation, and the art intervention showed significantly greater physiological relaxation, indicated by an increase in HRV parameters. This study also shows the effectiveness of HRV for measuring physiological arousal status [22].

We have rewritten the conclusion section as follows:

[Specifically, the observed lower Low Frequency (LF) power in patients with tic disorders suggests a potential chronic stress state, aligning with previous research that indicates chronic stress can lead to diminished autonomic flexibility.] -> A key finding of our study is that adolescents with tic disorders exhibited significantly lower LF power than the controls, suggesting an altered autonomic response pattern. Another key result was the positive correlation between SDNN, RMSSD, and various KIDSCREEN dimensions in the patient group, particularly autonomy and parent relations (Dimension 5). This suggests that greater HRV, indicative of better autonomic regulation, is linked to improved quality of life and psychological resilience in tic disorder patients. Conversely, a negative correlation between these HRV indices and quality of life was observed in the control group. This discrepancy underscores a fundamental difference in autonomic regulation between individuals with tic disorders and typically developing controls.

[The correlations between HRV parameters, particularly SDNN and RMSSD, and various dimensions of quality of life in the patient group highlight the intricate relationship between physiological stress responses, tic symptoms, and overall quality of life. Our study advocates for integrating stress management strategies and interventions to improve autonomic regulation in treating tic disorders. Such approaches could potentially enhance the quality of life for adolescents suffering from these conditions.]->

These results collectively highlight the intricate interplay between autonomic function, stress adaptation, and quality of life in tic disorder patients. The observed differences in HRV patterns suggest that tic disorders are not merely a neuropsychiatric condition but also involve physiological dysregulation that warrants further investigation. Future studies should explore whether targeted interventions, such as stress management techniques and HRV-based training, can improve autonomic function and reduce tic burden. In light of these findings, our study strongly advocates for a more integrative approach to tic disorder management, considering both psychological and physiological factors. The observed autonomic dysregulation suggests that incorporating HRV assessments and stress regulation strategies may enhance treatment outcomes and improve adolescents' overall well-being.

Deleted -> [In conclusion, this research contributes to a deeper understanding of the psychophysiological aspects of tic disorders. It highlights the critical role of stress and autonomic nervous system functioning in these conditions. Future studies should aim to explore the long-term effects of stress management and autonomic regulation interventions on tic disorders, potentially paving the way for more effective and comprehensive treatment modalities.]

Journal Requirements:

We have thoroughly checked our reference list and changed the description format regarding reference number 17. We have rewritten the information about the reference as follows.

[Camm AJ, Malik M, Bigger JT, Breithardt G, Cerutti S, Cohen RJ, et al. Heart rate variability: standards of measurement, physiological interpretation and clinical use. Task Force of the European Society of Cardiology and the North American Society of Pacing and Electrophysiology. Circulation. 1996;93(5):1043-65. ] -> Task Force of the European Society of Cardiology and the North American Society of Pacing and Electrophysiology. Heart rate variability: standards of measurement, physiological interpretation and clinical use. Circulation. 1996;93(5):1043-65.

---

## [Decision Letter · Decision Letter 2]

6 Mar 2025

The usefulness of heart rate variability in adolescents with tic disorder: Focused on interplay with quality of life

PONE-D-24-09801R2

Dear Dr. Lee,

We’re pleased to inform you that your manuscript has been judged scientifically suitable for publication and will be formally accepted for publication once it meets all outstanding technical requirements.

Kind regards,

Metha Chanda, D.V.M.,Ph.D., DTBVM

Academic Editor

PLOS ONE

---

## [Editor Report · Acceptance letter]

PONE-D-24-09801R2

PLOS ONE

Dear Dr. Lee,

I'm pleased to inform you that your manuscript has been deemed suitable for publication in PLOS ONE. Congratulations! Your manuscript is now being handed over to our production team.

Kind regards,

on behalf of

Associate Professor Metha Chanda

Academic Editor

PLOS ONE